# Suppression and facilitation of human neural responses

**Michael-Paul Schallmo[1]\*, Alexander M Kale[1], Rachel Millin[1], Anastasia V Flevaris[1], Zoran Brkanac[2], Richard AE Edden[3], Raphael A Bernier[2], Scott O Murray[1]**

[1]Department of Psychology, University of Washington, Seattle, United States; [2]Department of Psychiatry and Behavioral Sciences, University of Washington, Seattle, United States; [3]Department of Radiology and Radiological Science, Johns Hopkins University, Baltimore, United States

**Abstract** Efficient neural processing depends on regulating responses through suppression and facilitation of neural activity. Utilizing a well-known visual motion paradigm that evokes behavioral suppression and facilitation, and combining five different methodologies (behavioral psychophysics, computational modeling, functional MRI, pharmacology, and magnetic resonance spectroscopy), we provide evidence that challenges commonly held assumptions about the neural processes underlying suppression and facilitation. We show that: (1) both suppression and facilitation can emerge from a single, computational principle – divisive normalization; there is no need to invoke separate neural mechanisms, (2) neural suppression and facilitation in the motion-selective area MT mirror perception, but strong suppression also occurs in earlier visual areas, and (3) suppression is not primarily driven by GABA-mediated inhibition. Thus, while commonly used spatial suppression paradigms may provide insight into neural response magnitudes in visual areas, they should not be used to infer neural inhibition.

DOI: https://doi.org/10.7554/eLife.30334.001

**\*For correspondence:**
schal110@umn.edu

**Competing interests:** The authors declare that no competing interests exist.

## Introduction

Processes that regulate the level of activity within neural circuits (*Carandini and Heeger, 2012*) are thought to play a critical role in information processing by enabling efficient coding (*Vinje and Gallant, 2000*). Both suppression and facilitation of neural responses are well-known to emerge in the visual system via spatial context effects, and have a variety of perceptual consequences. For example, the perception of visual motion has been reliably shown to depend on the size and contrast of a stimulus (*Tadin et al., 2003*; *Tadin, 2015*). Specifically, more time is needed to discriminate the direction of motion of a large high-contrast grating compared to one that is small. This seemingly paradoxical effect is referred to as spatial suppression and has been suggested to reflect GABAergic inhibitory interactions from extra-classical receptive field (RF) surrounds (*Figure 1A and B*). The effect of size on duration thresholds is reversed for a low-contrast stimulus – less time is needed to discriminate motion direction for a large compared to small stimulus. This facilitation of behavior is referred to as spatial summation and has been suggested to reflect neural enhancement from RF surrounds (e.g. glutamatergic excitation) and/or an enlargement of RFs at low contrast (*Figure 1C*).

Strong assumptions are often made about the neural processes underlying these seemingly complex interactions between size and contrast during motion perception. This paradigm has been applied to the study of multiple clinical phenomena including schizophrenia (*Tadin et al., 2006*), major depressive disorder (*Golomb et al., 2009*), migraine (*Battista et al., 2010*), autism spectrum disorder (*Foss-Feig et al., 2013*; *Rosenberg et al., 2015*; *Sysoeva et al., 2017*), epilepsy (*Yazdani et al., 2017*), and Alzheimer's disease (*Zhuang et al., 2017*), as well as normal aging (*Betts et al., 2005*) and ethanol intoxication (*Read et al., 2015*). Conclusions about how neural

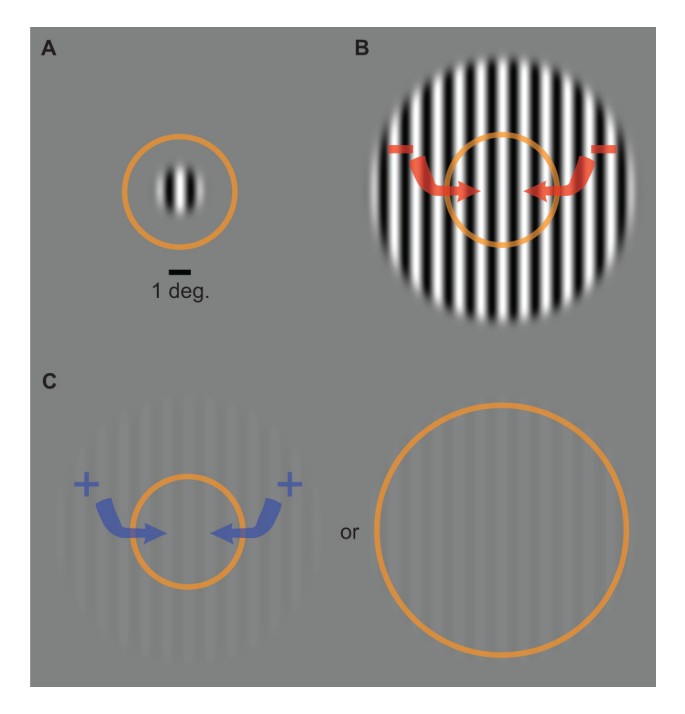

**Figure 1.** Common assumptions. The direction of motion of a small stimulus (**A**); contrast = 98%, diameter = 2°) can be perceived after a shorter presentation duration than a larger stimulus (**B**); diameter = 12°). This has been suggested to reflect the inhibitory influence of the extra-classical RF surround (red arrows) in motion-sensitive neurons in MT. Suppression turns to facilitation at low contrast (**C**; 3%), which has been assumed to reflect excitation from the surround and/or expansion of the classical RF. Orange ring represents the size of RFs in the foveal region of MT as measured in macaques (*Raiguel et al., 1995*; *Liu et al., 2016*). Comparable RF sizes in human MT are assumed (*Tadin et al., 2003*; *Amano et al., 2009*).
DOI: https://doi.org/10.7554/eLife.30334.002

processing is altered in these conditions have been drawn based on differences in duration thresholds relative to those of control observers. Generally, it has been assumed that spatial suppression and summation reflect distinct neural mechanisms that rely on inhibitory and excitatory processes, respectively (*Ma et al., 2010*; *Yoon et al., 2010*; *Cook et al., 2016*; *Haider et al., 2010*; *Adesnik et al., 2012*; *Nienborg et al., 2013*; but see [*Ozeki et al., 2004*; *Ozeki et al., 2009*; *Shushruth et al., 2012*; *Sato et al., 2016*; *Liu and Pack, 2014*), within brain regions involved in visual motion processing (particularly area MT).

Here, we test these assumptions directly and show that: (1) spatial suppression and summation in fact naturally emerge from a single, well-established neural computation observed in visual cortex – divisive normalization (*Heeger, 1992*; *Reynolds and Heeger, 2009*); there is no need to posit separate mechanisms. (2) While neural responses in human MT complex (hMT+) indexed with fMRI correspond well with the measured perceptual effect, there is substantial suppression in earlier visual areas; thus, it is possible that hMT+ 'inherits' suppression from earlier stages of processing. (3) Two separate methodologies – magnetic resonance spectroscopy (MRS) and pharmacological potentiation of GABA$_A$ receptors – fail to show a direct link between spatial suppression and the strength of neural inhibition. Although we find that inhibition plays a role in motion perception, increases in duration threshold as a function of stimulus size should not be taken as an index of inhibitory processing. In total, our results suggest that a single computational principle – divisive normalization – can account for spatial context effects and that suppressive context effects are not driven by neural inhibition.

## Results

### Quantifying behavior

To quantify spatial suppression and summation psychophysically, we measured motion duration thresholds (see Materials and methods) for 10 subjects in each of six different stimulus conditions, with sinusoidal luminance gratings at three different sizes (small [s], medium [m], and big [b]; diameter = 1, 2 and 12°, respectively) and two different contrasts (low = 3%; high = 98%; *Figure 2A–C*). The effect of stimulus size was quantified using a size index (SI; computed using the difference in thresholds between small and larger size conditions; see Materials and methods, *Equation 2*). Negative SI values indicate more time was needed for motion discrimination with larger stimuli (spatial suppression), while positive values indicate shorter durations for larger stimuli (spatial summation). As expected (*Tadin et al., 2003*; *Tadin, 2015*), SIs depended on both size and contrast ($F_{2,9} = 27.3$, p=4 × 10$^{-6}$), with spatial suppression observed at high contrast and spatial summation at low contrast (*Figure 2D*).

### A single computational framework for suppression and summation

The psychophysical effects of spatial suppression and summation – sometimes attributed to distinct neural mechanisms (*Figure 1*) – appear to depend on a complex interplay between stimulus size and contrast. We examined whether this apparent complexity could be explained by a simple, well-established model of early visual cortical responses that incorporates divisive normalization (*Heeger, 1992*; *Reynolds and Heeger, 2009*), which can be summarized as:

$$R = \frac{E}{S + \sigma} \tag{1}$$

This model describes the response (*R*) to a visual stimulus in terms of an excitatory drive term (*E*;

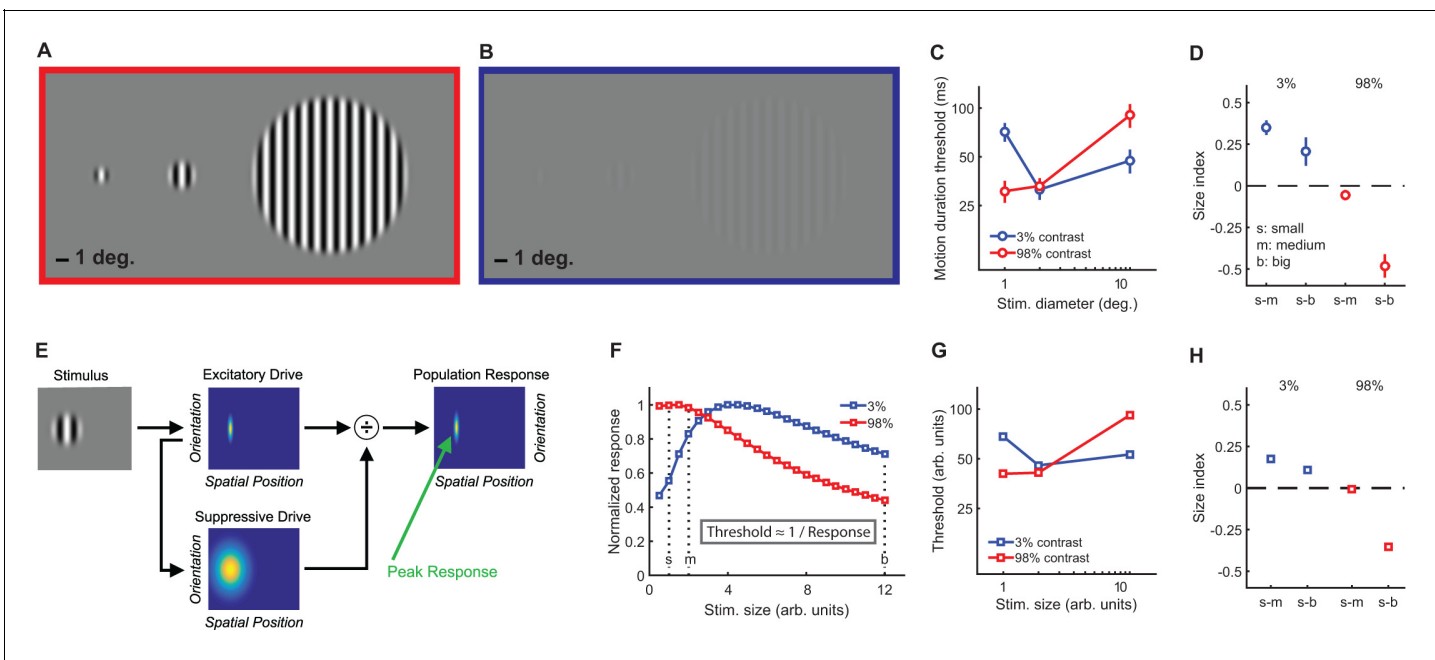

**Figure 2.** Stimuli, psychophysical results, and modeling. Small, medium, and big stimuli at high (**A**) and low contrast (**B**). The amount of time required to discriminate left- vs. right-moving stimuli with 80% accuracy (threshold in ms) is shown in (**C**) (average across N = 10 subjects, error bars are mean ± s. e.m.). Size indices (**D**) show the effect of increasing stimulus size, where negative values indicate that thresholds increase (suppression) and positive values indicate decreased thresholds (summation). A schematic representation of the normalization model is presented in (**E**) (for full model details, see Appendix 1), with the peak predicted responses for different stimulus sizes and contrasts shown in (**F**) (responses for both contrasts normalized to a maximum value of 1). As noted in the inset, predicted thresholds for motion discrimination are inversely proportional to these peak responses. Thresholds (**G**) and size indices (**H**) predicted by the model show a good qualitative match to the psychophysical data (**C** and **D**).
DOI: https://doi.org/10.7554/eLife.30334.003

reflecting the strength of the input, that is, stimulus contrast), divided by the sum of a suppressive drive term ($S$; also depends on input strength but is spatially broader; *Figure 2E*) plus a small number known as the semi-saturation constant ($\sigma$; controls response sensitivity; see Materials and methods and Appendix 1 for further model details). This model has been used previously to describe how normalization may contribute to a wide range of neural functions, from early visual processing and attention to olfaction and decision making (*Carandini and Heeger, 2012*). Additional model parameters that determine spatial sensitivity (e.g. excitatory spatial pooling width; *Appendix 1—table 1*) were chosen to match the physiological properties of neurons in visual cortex that are thought to drive spatial suppression during behavior (*Tadin et al., 2003*; *Amano et al., 2009*). We found that a good qualitative match between our psychophysical motion discrimination data and the model predictions (*Figure 2F–H*) could be obtained by assuming that the amount of time required for motion discrimination is inversely related to the peak modeled response (see Materials and methods; *Equation 3*).

At high contrast, as stimulus size grows the modeled suppressive drive increases relative to the excitatory drive, so the predicted response decreases (i.e. spatial suppression; red curve in *Figure 2F*). This pattern generally holds as long as the spatial tuning width of the excitatory drive is comparable with the smaller stimulus sizes, and the suppressive spatial tuning is larger than this. For low contrasts and small sizes, the suppressive drive is relatively weak compared to the semi-saturation constant ($\sigma$), so the response to low-contrast stimuli increases with stimulus size until the suppressive drive is relatively larger than $\sigma$ (i.e. spatial summation; blue curve in *Figure 2F*). Note that the parameters that determine spatial selectivity (i.e. model receptive field size) do not vary with stimulus contrast. Instead, within the computational framework of this model, spatial suppression vs. summation depends on the strength of the suppressive drive relative to $\sigma$ at a given stimulus contrast. In general, given comparable spatial parameters to those used here, summation is predicted for values of $\sigma$ that are within about 2 orders of magnitude of the stimulus contrast, for which the suppressive drive is relatively small compared to $\sigma$. In macaque visual cortex, the peak neural response is likewise often observed at larger sizes for lower contrast stimuli (*Angelucci and Bressloff, 2006*; *Pack et al., 2005*). Thus, the complex interplay between stimulus contrast and size that results in spatial suppression and summation during motion perception may be accounted for by a simple divisive normalization rule – it is not necessary to invoke distinct neural mechanisms to explain both phenomena.

## Neural responses reflect suppression and summation

Both our modeling work and previous studies (*Liu et al., 2016*; *Tadin, 2015*; *Turkozer et al., 2016*) suggest that spatial suppression and summation depend on reduced and enhanced neural responses (respectively) within visual cortex, particularly in area MT. To test this hypothesis directly in human visual cortex, we measured fMRI responses to low- and high-contrast moving gratings at different sizes with the same 10 subjects from the psychophysical experiment above. In a blocked experimental design (*Figure 3A*) (*Zenger-Landolt and Heeger, 2003*; *Williams et al., 2003*), we measured the change in the fMRI response evoked by increasing stimulus size. We report the responses from two areas: (1) early visual cortex (EVC; *Figure 3—figure supplement 1A*) which was located at the foveal confluence of areas V1, V2, and V3, near the occipital pole and (2) human MT complex (hMT+; *Figure 3—figure supplement 1C*). For both EVC and hMT+, we used an independent localizer scan to define sub regions-of-interest (sub-ROIs) corresponding to cortical areas that selectively respond to the smallest stimulus size (*Olman et al., 2007*; *Schallmo et al., 2016*). During a single experimental scan, two stimulus sizes were presented in alternating 10 s blocks: either small and medium or small and big. Thus, the sub-ROIs – defined by the smallest stimulus size – were constantly stimulated during the scan. This method allowed us to directly compare the relative fMRI responses to larger vs. smaller stimuli (i.e. the response during the small block served as a baseline for the larger block, which was the condition of interest). If responses in EVC and hMT+ sub-ROIs did not change between the small and larger stimulus blocks (the null hypothesis), this would indicate no influence of surrounding regions on response magnitude. Increased responses would reflect spatial summation, while spatial suppression would yield response decreases.

In EVC, there was no evidence for neural summation at low contrast; responses did not change (small-to-medium [s-m]; *Figure 3B*; *Figure 3—figure supplement 1B*) or decreased slightly (s-b) when low-contrast stimuli became larger. However, at high contrast, suppression was observed in

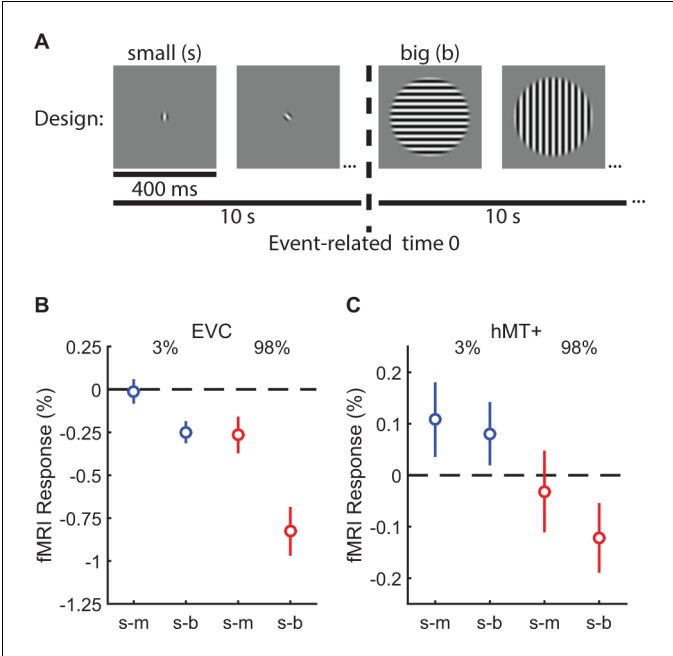

**Figure 3.** Measuring suppression and summation using functional MRI. This experiment measured the response to increasing stimulus size within regions of visual cortex representing the smallest stimulus. ROIs were localized in N = 10 subjects in EVC and N = 8 in hMT+. The blocked experimental design is illustrated in (**A**). Drifting gratings (400 ms on, 225 ms blank) of a particular size were presented within 10 s blocks. In (**B**), we show the change in the fMRI response within EVC and hMT + following the increase in stimulus size from small to medium (**s–m**) or small to big (**s–b**). The response to low-contrast stimuli (3%) is shown in blue, high contrast (98%) in red. Error bars are mean ± s.e.m. .

DOI: https://doi.org/10.7554/eLife.30334.004

The following figure supplement is available for figure 3:

**Figure supplement 1.** Regions-of-interest (ROIs) and response time courses for size-dependent fMRI responses.
DOI: https://doi.org/10.7554/eLife.30334.005

---

EVC ($F_{1,9}$ = 18.0, p=0.002) and was particularly strong in the small-to-big condition. In hMT+, there was evidence for both neural summation and suppression at low and high contrast, respectively. FMRI responses in hMT+ increased with stimulus size at low contrast (consistent with neural summation), and decreased at high contrast (consistent with neural suppression; *Figure 3C*; *Figure 3—figure supplement 1D*; $F_{1,7}$ = 9.0, p=0.020). This pattern of fMRI responses is a better match to the spatial summation and suppression observed using psychophysics (from *Figure 2D*), as compared to EVC which did not show any summation. The overall smaller fMRI response modulation in hMT+ vs. EVC was expected due to larger receptive fields in hMT+ (*Amano et al., 2009*), which reduce the retinotopic selectivity of the hemodynamic response within this ROI. These fMRI results are consistent with the proposal that increased and decreased neural activity within hMT+ contributes to spatial summation and suppression (respectively) during motion perception (*Liu et al., 2016*; *Tadin, 2015*). Observing both suppression and summation together within a single region is consistent with the framework of the normalization model (*Figure 2H*). In addition, it should be emphasized that suppression is clearly strong within earlier visual areas (EVC), even for low-contrast stimuli.

## The relationship between spatial suppression and GABA-mediated inhibition

After finding a match between neural responses in visual cortex and behavioral performance in this paradigm, we asked whether spatial suppression might be driven directly by GABAergic inhibition. Here, we used two separate methodologies: pharmacological potentiation of GABA_A receptors with the benzodiazepine lorazepam and measurements of individual differences in GABA concentration with magnetic resonance spectroscopy (MRS). Our *a priori* hypothesis was that if suppression

depends on GABA-mediated inhibition, increases in GABA signaling (either pharmacological or by natural variation in GABA concentration across individuals) should correspond with increased duration thresholds specifically for big stimuli – where suppression is greatest. We did not have a strong *a priori* hypothesis about how GABA signaling might affect duration thresholds for small stimuli, where suppression is minimal; however, our general intuition was that duration thresholds for small stimuli would be little affected.

## Pharmacologically enhanced inhibition

We first examined the effect of potentiating inhibition through administration of the benzodiazepine drug lorazepam, which acts as a positive allosteric modulator at the GABA$_A$ receptor (*Haefely, 1983*). This compound causes chloride channels to open more readily in response to GABA binding, which yields greater inhibition of neural action potentials. In a double-blind, placebo-controlled crossover experiment, 15 subjects received either a 1.5 mg dose of lorazepam or placebo during separate experimental sessions on different days (order randomized and counterbalanced across subjects). Participants then completed the psychophysical paradigm used to examine spatial suppression and summation. This allowed us to test the hypothesis that strengthening inhibition would lead to increased spatial suppression during motion perception.

We found that increasing inhibition via lorazepam did not lead to stronger suppression – in fact, we observed weaker spatial suppression (SIs were less negative overall) under lorazepam vs. placebo

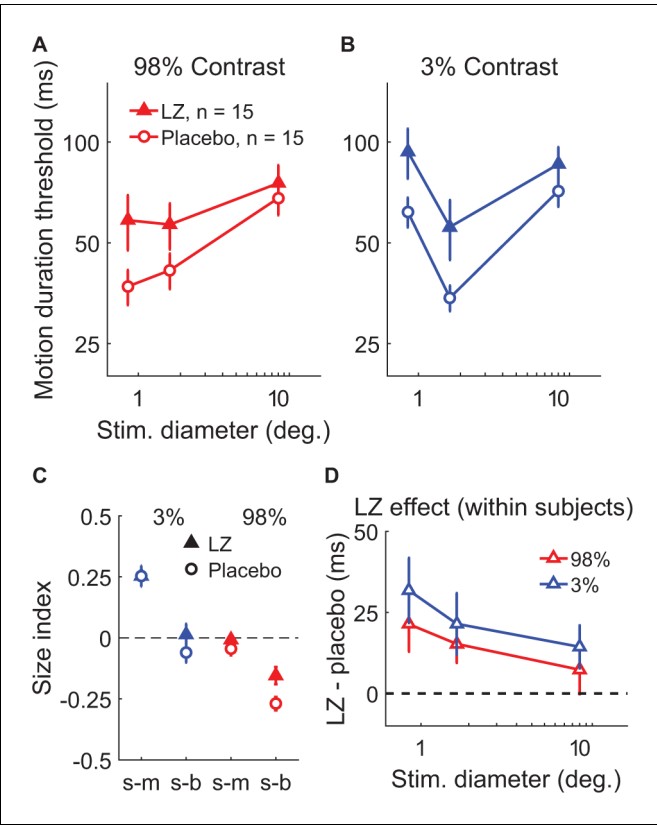

**Figure 4.** The effect of lorazepam on spatial suppression. Fifteen subjects took lorazepam (LZ) or placebo in a double-blind, crossover experiment. Duration thresholds were measured for high- (**A**) and low-contrast (**B**) moving gratings in each session. Size indices (**C**) and within-subjects effects of the drug (**D**) were calculated. Error bars are mean ± s.e.m.
DOI: https://doi.org/10.7554/eLife.30334.006
The following figure supplement is available for figure 4:

**Figure supplement 1.** Lorazepam model.
DOI: https://doi.org/10.7554/eLife.30334.007

(*Figure 4C*; $F_{1,14}$ = 6.51, p=0.023). Rather than increasing suppression, we found that lorazepam affected motion discrimination in two ways: first, lorazepam increased thresholds to some extent across all conditions ($F_{1,14}$ = 9.83, p=0.007). More importantly, the effect of lorazepam (drug minus placebo) was stronger for smaller stimuli (*Figure 4D*; $F_{1,14}$ = 6.91, p=0.020) at both low and high contrast. By increasing thresholds for smaller stimuli, lorazepam effectively weakened spatial suppression (i.e. reduced the difference in thresholds between small and larger stimuli). Thus, we found that potentiating inhibition at the $GABA_A$ receptor via lorazepam decreased, rather than increased spatial suppression in this paradigm. This result is not consistent with the idea that spatial suppression is directly mediated by neural inhibition.

## Measuring GABA via spectroscopy

To further examine the relationship between spatial suppression and inhibition, we measured the concentration of GABA+ (GABA plus co-edited macromolecules) within specific regions of visual cortex using MRS. GABA+ was quantified within moderately sized voxels (27 cm³) centered on functionally identified hMT+ (left and right hemispheres measured in separate scans and averaged; *Figure 5—figure supplement 1A*), anatomically identified EVC (average of 2 measurements in the same region; *Figure 5—figure supplement 1B*), and a control voxel in anatomically identified parietal cortex (*Figure 5—figure supplement 1C*). MRS data were acquired at rest (i.e. subjects were not asked to perform a specific task). The same subjects also participated in separate psychophysical and fMRI experiments to measure spatial suppression, comparable to those described above (N = 22 complete data sets). A summary of these data is provided in *Figure 5—figure supplement 2*, as a reference for the following analyses. Unlike manipulating GABA pharmacologically, MRS measurements of GABA+ are thought to reflect stable, individual differences in the baseline concentration of this neurotransmitter (*Evans et al., 2010*). Measuring this trait using MRS allowed us to test the hypothesis that subjects with more GABA+ in visual cortex would show greater spatial suppression.

Behaviorally, the strength of spatial suppression did not depend on the concentration of GABA+ in hMT+; no significant correlations were found between hMT+ GABA+ and SIs (*Figure 5—figure supplement 3A–F*; all $|r_{20-25}|$<0.32, uncorrected p-values>0.14). Further, there were no significant associations between fMRI measurements of spatial suppression in hMT+ and the concentration of GABA+ in this region (*Figure 5—figure supplement 3G–H*; $|r_{19}|$ < 0.29, uncorrected p-values>0.20). To aid visualization, in *Figure 5A–C* we show the psychophysical data with subjects split into two groups based on the concentration of GABA+ within hMT+ (median split). In one condition, suppression was numerically greater among individuals with high vs. low GABA+ in hMT+ (s-b, 98% contrast; *Figure 5C*). However, this effect was not statistically significant (*Figure 5—figure supplement 3C*), and appeared driven by a difference in thresholds for small stimuli (*Figure 5A*). These data did not show the predicted pattern of higher thresholds with big stimuli (for which suppression is strongest) among those with greater GABA+. Instead, we made the surprising observation that more GABA+ in hMT+ correlated with overall better psychophysical performance (lower thresholds on average) during motion direction discrimination (*Figure 5D*; $r_{20}$ = −0.46, p=0.030, uncorrected). These results suggest that more GABA in hMT+ is associated with better motion discrimination in general but not with the strength of spatial suppression.

When examining GABA+ measured within other areas (EVC or parietal cortex), no significant relationships were found with spatial suppression or overall psychophysical thresholds, nor did GABA+ correlate significantly with spatial suppression measured in EVC using fMRI (all $|r_{19-25}|$<0.28, uncorrected p-values>0.21; *Figure 5—figure supplement 4* and *Figure 5—figure supplement 5*). These findings suggest that the relationship between motion discrimination performance and GABA+ is specific to hMT+, and not a more general (e.g. brain-wide) phenomenon. Together, our spectroscopy results show no evidence for the predicted association between stronger spatial suppression and greater GABA in visual cortex, as indexed by MRS. Instead, higher GABA+ in hMT+ was associated with better motion discrimination performance overall. While this result differs from the effect of lorazepam on motion discrimination described above, we note that MRS reflects individual differences in GABA+ within visual brain areas that are (presumably) at homeostasis, whereas the effects of lorazepam can be attributed to a transient pharmacological strengthening of inhibition specific to the $GABA_A$ receptor class.

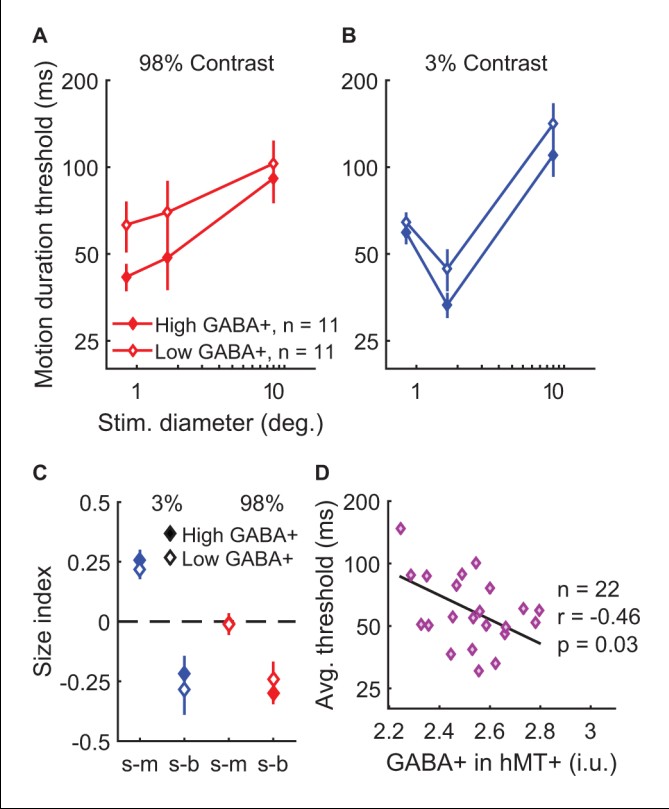

**Figure 5.** Examining task performance in terms of individual differences in GABA+ concentration in hMT+. To help illustrate the relationship between GABA+ measured in hMT+ and motion discrimination performance, thresholds (**A** and **B**) and size indices (**C**) are shown for subjects with lower (open symbols, N = 11) and higher GABA+ (filled symbols, N = 11; groups defined by median split). Error bars are mean ± s.e.m. As shown in (**D**), subjects with more GABA+ in hMT+ performed better overall during motion discrimination (lower average thresholds; geometric mean of all six stimulus conditions).

DOI: https://doi.org/10.7554/eLife.30334.008

The following figure supplements are available for figure 5:

**Figure supplement 1.** MRS voxel placement and GABA+ fitting.

DOI: https://doi.org/10.7554/eLife.30334.009

**Figure supplement 2.** Summary of psychophysical and fMRI data from the MRS experiment.

DOI: https://doi.org/10.7554/eLife.30334.010

**Figure supplement 3.** No significant relationship between GABA+ in hMT + and suppression or summation.

DOI: https://doi.org/10.7554/eLife.30334.011

**Figure supplement 4.** Examining task performance in terms of GABA+ in EVC.

DOI: https://doi.org/10.7554/eLife.30334.012

**Figure supplement 5.** Examining task performance in terms of GABA+ in parietal cortex.

DOI: https://doi.org/10.7554/eLife.30334.013

**Figure supplement 6.** MRS model.

DOI: https://doi.org/10.7554/eLife.30334.014

**Figure supplement 7.** FMRI contrast-response data.

DOI: https://doi.org/10.7554/eLife.30334.015

## Discussion

This study examined a number of common assumptions about the neural processes underlying spatial suppression and summation during motion perception. Although this paradigm has been posited as a behavioral index for inhibition, we did not find a major role for GABA in determining the strength of spatial suppression. While a few studies have probed the neurochemical underpinnings of spatial context processing in humans (*Read et al., 2015*; *Yoon et al., 2010*; *Cook et al., 2016*;

Song et al., 2017; van Loon et al., 2012), much of our current insight into the neural mechanisms of suppression and inhibition has come from work in animal models. Some reports suggest that GABAergic inhibition plays a direct role in surround suppression within visual cortex (Ma et al., 2010; Haider et al., 2010; Adesnik et al., 2012; Nienborg et al., 2013), while others have indicated that this suppression occurs via withdrawal of excitation (Ozeki et al., 2004; Ozeki et al., 2009; Shushruth et al., 2012; Sato et al., 2016), which may be balanced by reduced inhibition.

Here, we predicted that if suppression was directly mediated by GABA, then we would observe both stronger suppression with lorazepam vs. placebo, and a significant correlation between GABA MRS and suppression measures. However, this was not the case; lorazepam weakened suppression rather than strengthening it, and MRS measurements of GABA+ in visual cortex did not correlate significantly with suppression strength. Further, based on post-hoc power analyses (see Materials and methods), we conclude that there is no evidence for strong, or even moderate correlations (stronger than $r = 0.52$) between GABA+ and suppression in our data. Of course, it remains possible that we failed to detect a weaker correlation between GABA+ and spatial suppression. If such a subtle relationship exists ($r^2 < 0.3$; here undetected), this would perhaps suggest a more complex role for GABA in mediating spatial suppression than we first hypothesized. Nevertheless, our findings seem more consistent with the withdrawal of excitation as a mechanism for spatial suppression. This agrees with an earlier observation that blocking GABA in macaque MT had no effect on spatial suppression (Liu and Pack, 2014) (see also [Katzner et al., 2011]). However, our MRS findings appear to conflict with reports that greater mid-occipital GABA was associated with stronger surround suppression during contrast perception (Yoon et al., 2010; Cook et al., 2016). This discrepancy may be related to the methods for quantifying suppression, and/or differences in the role of inhibition during perception of contrast and motion.

Beyond spatial suppression, the results from our lorazepam and MRS experiments help to clarify the role of GABAergic inhibition in mediating motion perception. We considered how these results might be described using the normalization model. We found that potentiating inhibition at the $GABA_A$ receptor via lorazepam decreased spatial suppression, rather than strengthening it. Figure 4—figure supplement 1A–D shows how a reduction in both input (i.e. contrast) and output (i.e. response) gain yields predicted thresholds that mirror the observed effects of lorazepam (compare with Figure 4A–D; also see Appendix 1—table 1). Specifically, higher thresholds for smaller stimuli following lorazepam may be accounted for in the normalization model by reducing the strength of the input (i.e. reduced contrast gain). Lowering contrast gain raised the predicted thresholds for small stimuli but had little effect when stimuli were large. We therefore speculate that the effect of lorazepam in reducing spatial suppression may be consistent with reduced contrast gain within brain regions relevant to motion perception (e.g. MT).

From previous work that manipulated $GABA_A$ receptor function in animal models (Katzner et al., 2011; Thiele et al., 2004), it may be expected that potentiating inhibition at this receptor would reduce neural responsiveness overall (i.e. lower response gain). Beyond increasing thresholds for small stimuli, lorazepam also raised motion discrimination thresholds across conditions (Figure 4D), an effect that is consistent with overall lower neural responses. While reducing input gain in the model (as above) slightly reduces the predicted thresholds in all conditions, such an effect may also be modeled by reducing response gain (scaling down predicted responses; Figure 4—figure supplement 1A–D; Appendix 1—table 1). This is also consistent with earlier behavioral studies showing that benzodiazepine administration generally reduces visual performance (van Loon et al., 2012; Giersch and Herzog, 2004). However, we note that the observed effects of lorazepam were specific to threshold-level motion discrimination (especially for smaller stimuli); catch trial performance was not affected (see Materials and methods), which argues against more general pharmacological effects (e.g. fatigue). In summary, while the effect of lorazepam during motion discrimination may be consistent with weaker contrast gain (and perhaps also lower response gain), our findings do not support the idea that spatial suppression is directly mediated by inhibition.

We additionally showed that higher baseline GABA (as measured by MRS) is associated with better, rather than suppressed motion perception. Because this correlation was unexpected, and no statistical correction for multiple comparisons was performed, this result should be interpreted with caution. To understand why more GABA in hMT+ might lead to better motion discrimination, we considered two possibilities. First, better performance (lower thresholds) might result if GABA increased neural activity in this region – this proposal does not seem parsimonious, given the well-established inhibitory function of GABA in the adult nervous system (McCormick, 1989). Alternatively, if more baseline GABA lowers the behavioral response criterion (without necessarily changing the neural

response), possibly by improving neural signal-to-noise (*Leventhal et al., 2003*; *Edden et al., 2009*; *Puts et al., 2011*), then we would expect better performance with higher GABA in hMT+. Indeed, within our model (*Equation 3*), reducing the response criterion has the same effect on the predicted threshold as an increased neural response. In *Figure 5—figure supplement 6A–C*, we show how the effect of changing criteria can be described in terms of the normalization model (compare with *Figure 5A–C*; see also *Appendix 1—table 1*). Note that changing the criterion has no effect on the strength of spatial suppression predicted by the model (*Figure 5—figure supplement 6C*). This is consistent with our finding that the concentration of GABA+ in hMT+ does not influence spatial suppression measured psychophysically (*Figure 5C*; *Figure 5—figure supplement 3A–F*) or with fMRI (*Figure 5—figure supplement 3G–H*). In terms of the motion discrimination task, a lower response criterion indicates that less sensory evidence (i.e. shorter stimulus duration) is used to reach the same decision (i.e. left- vs. right-ward motion) (*Huk and Shadlen, 2005*). This would lead to more-rapid perceptual decision making, and thus better performance. Thus, lower response criteria with higher baseline GABA may plausibly account for our MRS data. As with our experiment using lorazepam, these MRS data contradict the notion that GABAergic inhibition directly determines the strength of spatial suppression. Altogether, our findings suggest that the assumption of a direct link between spatial suppression and inhibition is invalid.

We considered whether spatial suppression and summation might be described in terms of a single computation – divisive normalization – a model for early visual processing in which a neuron's response is suppressed (divided) by the summed response of its neighbors (*Heeger, 1992*; *Reynolds and Heeger, 2009*). We are not the first to suggest that the normalization model may account for these phenomena; Rosenberg and colleagues (*Rosenberg et al., 2015*) used weaker normalization to explain superior motion discrimination performance in autism spectrum disorder (*Foss-Feig et al., 2013*). Our study provides the first direct application of this model to this paradigm in typically developing individuals under a variety of experimental conditions. While it is perhaps not surprising that this model performs well in this context (*Carandini and Heeger, 2012*), earlier work has modeled spatial suppression and summation in terms of other divisive (*Betts et al., 2012*) or subtractive (*Tadin and Lappin, 2005*) computations. Generally, these earlier models have treated 'excitatory' and 'suppressive' mechanisms separately (e.g. different contrast sensitivity). The normalization model (*Reynolds and Heeger, 2009*) used here provides a simpler account, wherein the suppressive drive is essentially the same as the excitatory drive, only broader in space and orientation (see Appendix 1). We found that this normalization model is sufficient to explain the complex interplay between stimulus size and contrast that produces psychophysical (or neural) suppression and summation. This framework appears more parsimonious than positing separate mechanisms (neural or computational) operating at different stimulus contrasts to produce suppression and summation.

Our modeling work shows that normalization can describe spatial suppression and summation under a variety of different experimental conditions. We found good agreement between the predicted model response (*Figure 2G and H*), psychophysics (*Figure 2C and D*), and the fMRI response in hMT+ (*Figure 3C*). That both suppression and summation were observed in the fMRI response from a sub-region of hMT+ representing the stimulus center indicates that these two effects are co-localized in cortex, and both could plausibly occur within a single neural population (*Pack et al., 2005*), which is consistent with the proposed model framework. Further, the way in which spatial suppression was affected by lorazepam (*Figure 4—figure supplement 1A–D*), but not baseline GABA in hMT+ (*Figure 5—figure supplement 6A–C*), can be explained in terms of the normalization model (*Appendix 1-Table 1*). It is worth noting that the lorazepam data were described by an exponential reduction of contrast gain, which we speculate may reflect the compounded reduction of neural responses across multiple stages of visual processing (e.g. retina, lateral geniculate nucleus, visual cortex) due to systemic potentiation of inhibition at the $GABA_A$ receptor. Although recent work has suggested some limitations for describing spatial suppression in terms of normalization (*Heeger et al., 2017*), we find that this model framework is sufficiently general to describe the suppression effect across our different experiments.

Finally, we examined the extent to which suppression and summation may be attributed to modulation of neural activity within area MT. Our results generally supported this hypothesis; fMRI responses in hMT+ showed better agreement with spatial suppression and summation measured psychophysically, as compared with the responses in EVC (*Figure 3B and C*). Further, better motion

discrimination across individuals was predicted by higher GABA+ in hMT+ (*Figure 5D*), but not in EVC (*Figure 5—figure supplement 4D*). The idea that spatial suppression during perception is driven (at least in part) by surround suppression within MT is consistent with recent work in primate models (*Liu et al., 2016*). For many neurons in primate MT, the presence of stimuli within the extra-classical receptive field surround suppresses (or enhances) the response to stimuli within the receptive field center (*Liu et al., 2016*; *Pack et al., 2005*; *Born and Tootell, 1992*; *Born, 2000*). The proposed link between perceptual suppression and neural suppression in MT has received further (if limited) support from studies in humans (*Turkozer et al., 2016*; *Tadin et al., 2011*). Using fMRI, MRS, and modeling, our findings extend this link by showing a correspondence between neural processing in hMT+ and motion discrimination performance in human subjects.

While neural processing in hMT+ seemed more closely linked to perception, strong suppression of fMRI responses was observed in EVC at both low and high stimulus contrast (*Figure 3B*). Feed-forward and feedback connections between EVC and MT are thought to play an important role in spatial context processing (*Angelucci and Bressloff, 2006*); our data alone are not sufficient to determine the extent to which MT inherits suppression from EVC and/or drives this suppression via feedback. However, surround suppression in early visual areas (e.g. V1, V2) is well established in animal models (*Angelucci and Bressloff, 2006*), and suppressed fMRI responses in these areas generally correspond with perceptual suppression during contrast judgments (*Zenger-Landolt and Heeger, 2003*; *Schallmo et al., 2016*; *Joo et al., 2012*). This raises the possibility that some amount of spatial suppression during motion discrimination may be attributed to neural suppression in EVC. Other aspects of motion processing in MT are also believed to be inherited from earlier stages of visual processing (*Kohn and Movshon, 2003*; *Glasser et al., 2011*).

One issue for comparing fMRI results between hMT+ and EVC is the difference in RF sizes between areas. Neurons in foveal MT are thought to have a RF diameter of ~5°, while RFs in foveal EVC are much smaller (~1–2° diameter, depending on the visual area) (*Raiguel et al., 1995*; *Liu et al., 2016*; *Tadin et al., 2003*; *Amano et al., 2009*). Our largest stimuli (12° diameter) are therefore expected to extend into the extra-classical RF surround for foveal neurons in both areas. Our ability to measure spatial suppression and summation with fMRI will also be affected by RF size; larger RFs in MT should make these measurement more difficult, because of less retinotopic specificity within a given voxel (*Amano et al., 2009*). However, in about 80% of subjects, we were able to identify voxels within hMT+ that responded selectively to foveal > surrounding stimuli. These hMT+ sub-ROIs showed both spatial suppression and summation (*Figure 3*), indicating that we have the necessary spatial specificity to observe these phenomena in hMT+ in general. The fMRI responses to increasing stimulus size from voxels outside of the center-selective hMT+ sub-ROIs were very different (data not shown), and cannot be attributed primarily to spatial suppression or summation. Future work with higher spatial resolution imaging may provide greater insight into the nature of spatial suppression and summation within hMT+.

## Materials and methods

### Participants

A total of 48 adult human subjects were recruited across all experiments. First, 10 subjects participated in both the initial experiments characterizing spatial suppression psychophysically and with fMRI (eight males and two females, mean age 30 years, *SD* = 6.4 years). Second, 15 subjects (seven males and eight females, mean age 27 years, *SD* = 4.4 years) completed the lorazepam experiment. Four subjects participated in both of these first two sets of experiments. Third, 27 subjects (12 males and 15 females, mean age 24 years, *SD* = 3.6 years) completed the MRS experiment.

All subjects reported normal or corrected-to-normal vision and no neurological impairments. Before enrollment in the lorazepam experiment, subjects were screened for potential drug interactions (e.g. antidepressants, antipsychotics, sedatives, or stimulants), allergies to benzodiazepines, neurological impairments, and neurodevelopmental disorders. Subjects participating in MRS reported no psychotropic medication use, no more than one cigarette per day within the past 3 months, no illicit drug use within the past month, and no alcohol use within 3 days prior to scanning. Subjects provided written informed consent prior to participation and were compensated for their time. All experimental procedures were approved by the University of Washington Institutional

Review Board, and conformed to the ethical principles for research on human subjects from the Declaration of Helsinki.

## Visual display and stimuli

Psychophysical experiments were performed using one of two display apparatuses in different physical locations for logistical reasons: (1) a ViewSonic G90fB CRT monitor (refresh rate = 85 Hz; used for the data shown in *Figure 2*) or (2) a ViewSonic PF790 CRT monitor (120 Hz; used for all other experiments) with an associated Bits# stimulus processor (Cambridge Research Systems, Kent, UK). In both cases, stimuli were presented on Windows PCs in MATLAB (MathWorks, Natick, MA) using Psychtoolbox-3 (*Brainard, 1997*; *Pelli, 1997*; *Kleiner et al., 2007*), and a chin rest was used to stabilize head position. During fMRI, stimuli were displayed via projector; either an Epson Powerlite 7250 or an Eiki LCXL100A (following a hardware failure), both operating at 60 Hz. Images were presented on a semicircular screen at the back of the scanner bore, and viewed through a mirror mounted on the head coil. Stimuli during fMRI were displayed using Presentation software (Neurobehavioral Systems, Berkeley, CA). The luminance of all displays was linearized. Viewing distance was 52 cm for psychophysical display #1, and 66 cm for both display #2 and in the scanner.

In each experiment, we presented drifting sinusoidal luminance modulated gratings at two different Michelson contrast levels (low = 3%, high = 98%) and three different sizes (small, medium and big; see *Figure 2A and B*), following the method of Foss-Feig and colleagues (*Foss-Feig et al., 2013*). Stimulus diameter was 1, 2 and 12° visual angle for the small, medium, and big stimuli (respectively) in all fMRI experiments and the first psychophysical experiment (data shown in *Figure 2*; using display #1). Due to a coding error, the stimulus diameter was slightly smaller in all subsequent psychophysical experiments (performed using display #2; diameter = 0.84, 1.7 and 10°). Drift rate was always four cycles/s. Stimuli were presented centrally on a mean luminance background, and had a spatial frequency of 1 cycle/° (display #1 and fMRI) or 1.2 cycles/° (display #2). Gratings were presented within a circular window, whose edges were blurred with a Gaussian envelope ($SD$ = 0.25° for display #1 and fMRI, 0.21° for display #2). Gratings were presented within a circular window, whose edges were blurred with a Gaussian envelope. Note that this spatial envelope differs from the standard Gaussian envelope used previously (*Tadin et al., 2003*), and yielded higher average contrast and a sharper edge profile than comparable Gaussian windows; both of these factors likely influenced the precise pattern of suppression and summation (*Tadin, 2015*) observed here.

## Paradigm and data analysis

### Psychophysics

Subjects were asked to determine whether a briefly presented vertical grating drifted left or right (randomized and counterbalanced). Trials began with a central fixation mark; either a small shrinking circle (850 ms, for the MRS experiments) or a static square (400 ms, all other experiments). This was followed by a blank screen (150 ms), after which the grating stimuli appeared (variable duration, range 6.7–333 ms), followed by another blank screen (150 ms), and finally a fixation mark (the response cue). Subjects indicated their response (left or right) using the arrow keys. Response time was not limited. To permit very brief stimulus presentations, gratings appeared within a trapezoidal temporal envelope, following an established method (*Foss-Feig et al., 2013*). Thus, the first and last frames were presented at sub-maximal contrast, and the duration was defined by the full-width at half-maximum contrast.

Duration of the grating stimuli varied across trials according to a Psi adaptive staircase procedure (*Kingdom and Prins, 2010*) controlled using the Palamedes toolbox (*Prins and Kingdom, 2009*). Duration was adjusted across trials based on task performance, to determine the amount of time needed to correctly discriminate motion direction with 80% accuracy (i.e. the threshold duration). Staircases were run separately to determine thresholds for each of the six stimulus conditions (two contrasts x three sizes, as above). Condition order was randomized across trials. Thirty trials were run per staircase within a single run (approximately 6 min). There were also 10 catch trials per run (all big, high-contrast gratings, 333 ms duration), which were used to assess off-task performance. Each subject completed four runs, with a total experiment duration of about 30 min. Example and

practice trials were presented before the first run. For five subjects in the MRS experiment, thresholds were not obtained for the smallest stimulus size.

Psychometric thresholds and slopes were quantified for each run by fitting the discrimination accuracy data with a Weibull function using maximum likelihood estimation (*Kingdom and Prins, 2010*). Guess and lapse rate were fixed at 50% & 4%, respectively. Threshold duration was defined at 80% accuracy based on this fit. Threshold estimates below 0 ms or above 500 ms were excluded; a total of 4 such threshold estimates were excluded across all experiments. When averaging across thresholds from different stimulus conditions (e.g. *Figure 5D*) we computed a geometric mean, to account for the fact that the threshold range varied across conditions. The effect of stimulus size on task performance was quantified using a standard size index (SI) metric (*Foss-Feig et al., 2013*), such that:

$$SI = log_{10}(smaller\ threshold) - log_{10}(larger\ threshold) \qquad (2)$$

## Computational modeling

We examined the extent to which spatial suppression could be qualitatively explained in terms of a well-established divisive normalization model (*Heeger, 1992*; *Reynolds and Heeger, 2009*). An equation that summarizes the model (*Equation 1*) is included in the Results, and a graphical depiction of the model is presented in *Figure 2E*. Full details of the model are provided in Appendix 1. Briefly, the values of *E* and *S* in *Equation 1* depend on the properties of the stimulus (e.g. size, orientation, contrast), as well as a number of other parameters that determine how sensitive the model is to different stimulus properties (e.g. the spatial extent of excitation and suppression; *Equations A1 and A2*). The value of $\sigma$ determines the sensitivity of the response to weak stimuli, and prevents the response from being undefined when a stimulus is absent. These parameters are derived from an extensive literature describing how neurons in early visual cortex respond to different properties of visual stimuli (*Reynolds and Heeger, 2009*), and are listed for each instantiation of the model in *Appendix 1—table 1*.

A good qualitative match between the model and our psychophysical data was obtained with minimal adjustments of the model assumptions (i.e. free parameters were adjusted manually, rather than estimated based on a computational fit to our data). We used a 'winner-take-all' decision rule in reading out the population response from the model, which had the consequence of choosing the response from the center of the population being modeled (i.e. the response to the center of the stimulus, which was always the largest within the population). To equate the response predicted by the model to motion discrimination thresholds, we assumed an inverse relationship between predicted response and discrimination time, such that:

$$Threshold = Criterion\ /\ Response \qquad (3)$$

where *Threshold* is the amount of time needed to discriminate the direction of stimulus motion, *Criterion* represents an arbitrary response value that must be reached to make the perceptual judgment (*Huk and Shadlen, 2005*), and *Response* is the predicted response rate (from *Equation 1*). This framework is consistent both with previous modeling efforts (*Betts et al., 2012*; *Tadin and Lappin, 2005*), and electrophysiological work showing a close correspondence between psychophysical discrimination thresholds and neural response magnitudes in macaque MT during motion perception (*Liu et al., 2016*; *Britten et al., 1992*).

The value of *Criterion* was held constant across stimulus sizes for a given contrast, in order to facilitate direct comparisons of the effect of size on the modeled threshold. However, the *Criterion* value varied across different versions of the model (*Appendix 1—table 1*) and different contrast levels – typically, a lower *Criterion* was used to model thresholds for 3% versus 98% contrast stimuli. A lower *Criterion* indicates a difference in the process of accumulating sensory evidence to reach a perceptual decision, such that less evidence is required (*Huk and Shadlen, 2005*). Using lower *Criterion* values was important in cases where duration thresholds were equivalent or shorter for low- vs. high-contrast stimuli (e.g. *Figure 2C*, medium size). We chose to adopt this modeling approach after comparing motion discrimination thresholds for medium-sized stimuli with the fMRI responses to briefly presented moving gratings (2° diameter) at 3% and 98% contrast (vs. fixation alone; see below). We found that in both EVC and hMT+, fMRI responses were significantly higher for 98% vs. 3% contrast stimuli (*Figure 5—figure supplement 7*; $F_{1,26} = 113$, p<0.001), as expected. However,

we found duration thresholds for comparable stimuli were equal or lower for gratings at 3% vs. 98% across multiple experiments (*Figure 2*, *Figure 4* and *Figure 5*). This leads us to the conclusion is that there must be additional factor(s), beyond response magnitudes in hMT+ and/or EVC, that contribute to duration thresholds at low vs. high contrast. We used different *Criterion* values in the current model to account for such factors. We suggest that one plausible explanation for this phenomenon is stimulus onset masking (*Tadin, 2015*); masking has been shown to contribute to spatial suppression, and is thought to be stronger for high-contrast stimuli (*Churan et al., 2009*; *Tsui and Pack, 2011*).

We note that our intention with this modeling work was not to find the precise parameter values that provided the best algorithmic fit to our data. Rather, we sought to demonstrate that, using a reasonable set of manually derived parameters, a well-established model of spatial context processing (divisive normalization) is sufficient to explain both suppression and summation during motion discrimination. In general, we used parameter values that were similar to previous instantiations (*Reynolds and Heeger, 2009*; *Flevaris and Murray, 2015*), and/or approximate the realistic values of neurons in visual cortex. Rather than make any claims about the specifics of the parameter values, we instead note in the Results the relationships between parameters (e.g. suppressive drive having broader spatial tuning than excitatory drive) that are necessary to predict the general pattern of results from our experiments.

## Lorazepam

Lorazepam is a benzodiazepine that acts as a positive allosteric modulator at the $GABA_A$ receptor (*Haefely, 1983*). Rather that acting directly as an agonist (i.e. binding to the GABA receptor site), lorazepam binds to a separate 'benzodiazepine' site on the receptor, which increases the probability that the receptor's $Cl^-$ channel will open when GABA binds. This leads to stronger hyperpolarization of the postsynaptic neuron's membrane potential. Thus, the net effect of lorazepam is to potentiate inhibition at the $GABA_A$ receptor.

In separate experimental sessions separated by at least 1 week, subjects received either 1.5 mg lorazepam or placebo, with the order randomized and counter-balanced across subjects. The compounds were dispensed by a pharmacist who was not involved the study; both subjects and experimenters were blind to the order of drug and placebo until after both experimental sessions were complete. Following a 2 hr wash-in period, subjects completed the above psychophysical paradigm as part of a larger battery of experiments lasting approximately 1.5 hr. The order in which the spatial suppression paradigm was performed within this series was randomized and counter-balanced across subjects, but was always the same for the drug and placebo sessions within each subject. Instructions and practice trials were presented before the experiment in both sessions.

Catch trial accuracy was used to assess whether lorazepam affected cognitive performance in general, or motion perception more specifically. Accuracy was equivalently high in both placebo (mean = 99%, *SD* = 1.8%) and drug sessions (mean = 98%, *SD* = 3.6%; paired *t*-test, $t_{14}$ = 0.8, p=0.4), suggesting that lorazepam may have reduced threshold-level motion discrimination, but not task performance more generally (e.g. reduced performance due to fatigue).

## Functional MRI

Data were acquired on a Philips Achieva 3 Tesla scanner. A $T_1$-weighted structural MRI scan was acquired during each session with 1 mm isotropic resolution. Gradient echo fMRI data were acquired with 3 mm isotropic resolution in 30 oblique-axial slices separated by a 0.5 mm gap (2 s TR, 25 ms TE, 79° flip angle, A-P phase-encode direction). A single opposite direction (P-A) phase-encode scan was acquired for distortion compensation. Each scanning session lasted approximately 1 hr.

Our primary fMRI paradigms examined the change in the fMRI signal in response to an increase in stimulus size (e.g. spatial suppression). This involved presenting smaller and larger drifting gratings during alternating 10 s blocks (*Figure 3A*). This type of alternating-block design has been used previously to measure surround suppression during fMRI (*Zenger-Landolt and Heeger, 2003*; *Williams et al., 2003*) and was chosen for its simplicity and robustness to subject noise. For the data shown in *Figure 3* and *Figure 3—figure supplement 1*, stimulus diameter alternated between 1° and 2°, or 1° and 12° in separate 5 min runs. For those in *Figure 5—figure supplement 2*, *Figure 5—figure supplement 3*, and *Figure 5—figure supplement 4*, diameter alternated between 2° and

12°. Stimulus duration was 400 ms, with a 225 ms inter-stimulus interval. Sixteen gratings were presented at the center of the screen during each block; to prevent adaptation, gratings moved in one of eight possible directions in a randomized and counter-balanced order. Twenty-five blocks (13 small, 12 big) were presented during each run (125 TRs total). Stimulus contrast was either 3% or 98% (in separate runs). Each subject completed 2–4 runs at each contrast level.

Subjects performed a color/shape detection task at fixation during all fMRI experiments, in order to both control spatial attention and to minimize eye movements away from fixation. We note that this fixation task necessarily presented different attentional demands than our psychophysical paradigm (motion discrimination outside the scanner). Attention has been shown to affect center-surround interactions in some cases (*Flevaris and Murray, 2015*; *Zenger et al., 2000*) (but see (*Schallmo et al., 2016*; *Millin et al., 2014*; *Poltoratski et al., 2017*)). Here, keeping attention focused tightly at fixation during fMRI had the advantage of mimicking the manner in which spatial attention is thought to be deployed during motion direction discrimination for very short duration stimuli (*Liu et al., 2016*; *Tadin, 2015*).

For the data presented in *Figure 5—figure supplement 7*, stimulus contrast varied across blocks, rather than size. These scans began with a blank block in which only the fixation mark was presented on a mean gray background. Blocks of drifting gratings (2° diameter) at 3% and 98% contrast were then presented centrally in an alternating order, each followed by a blank block, to permit the fMRI response to return to baseline (6 low-contrast blocks, 6 high contrast, and 13 blank per run). Other stimulus parameters matched those from the fMRI experiments above. Subjects each completed 2–4 runs of the contrast experiment.

Functional localizer scans were acquired to facilitate ROI definition. Three different localizers were used; paradigm structure matched those above, except where noted. The first localizer was designed to identify human MT complex (hMT+); we use this notation to clarify that we did not attempt to distinguish areas MT and MST, both of which are motion selective (*Huk et al., 2002*). Drifting and static gratings (15% contrast) were presented centrally in alternating 10 s blocks (125 TRs total). Stimulus diameter was 1° (hMT+ data from *Figure 3* and *Figure 3—figure supplement 1*) or 2° (*Figure 5—figure supplement 2* & *Figure 5—figure supplement 3*). The second localizer was used to identify regions of visual cortex that represented the smallest stimulus size (*Olman et al., 2007*); checkerboard stimuli (100% contrast, phase-reversing at 8 Hz) were presented in center and surrounding regions in an alternating order across 16 blocks (10 s each, 80 TRs total). Center diameter was either 1° (with a 1° gap; *Figure 3* and *Figure 3—figure supplement 1*) or 2° (*Figure 5—figure supplement 2*, *Figure 5—figure supplement 3*, and *Figure 5—figure supplement 4*); inner and outer diameter of the surround annulus were always 2° and 12°, respectively. The third localizer was used to identify EVC ROIs for the contrast experiment (*Figure 5—figure supplement 7*); checkerboard stimuli (2° diameter, other parameters as in localizer #2) and blank backgrounds were presented in alternating blocks.

FMRI data were processed in BrainVoyager (Brain Innovation, Maastricht, The Netherlands), including motion and distortion correction, high-pass filtering (cutoff = 2 cycles/scan), and anatomical alignment. No spatial smoothing or normalization were performed; all analyses were within-subjects and ROI-based. ROIs were identified from the localizer data using correlational analyses (*Schallmo et al., 2016*), with an initial threshold of p<0.05 (Bonferroni corrected). ROIs were defined for each hemisphere in two anatomical regions: motion-selective hMT+ in the lateral occipital lobe (*Figure 3—figure supplement 1C*), and the region of EVC selective for the retinotopic position of the center stimulus (near the occipital pole; *Figure 3—figure supplement 1A*). ROI position was verified by visualization on an inflated model of the cortical white matter surface. The top 20 most-significant voxels (in functional space) within each hemisphere were selected for analysis. In a few cases, there were not 20 functional voxels within a hemisphere ROI that met the statistical threshold above. For these subjects, the threshold was relaxed until 20 voxels from the surrounding region were included. ROIs in all subjects satisfied an uncorrected, one-tailed significance threshold of p<0.002. Center-selective hMT+ sub-ROIs were defined based on a correlation analysis of their time course data during the center-vs.-surround localizer scan, with a further inclusion criterion of p<0.05 (one-tailed). This allowed us to examine the fMRI response to increasing stimulus size within hMT+ voxels that showed some selectivity for the retinotopic position of the center stimulus, in addition to significant motion selectivity. Sub-ROIs in hMT+ could not be identified in two subjects from the first fMRI experiment, and six subjects from the MRS and fMRI experiment.

Average time courses were extracted from each sub-ROI for further analyses in MATLAB using BVQXTools. Time course data were split into epochs spanning from 4 s before the start to 2 s after the end of each block of interest. The response baseline was determined by averaging the signal across all such epochs between 0 and 4 s prior to block onset. For the main fMRI experiment (e.g. *Figure 3*), we sought to measure suppression or summation in response to an increase in stimulus size. Thus, the block of interest contained the larger stimuli, and the response from the end of the preceding block of smaller stimuli served as the baseline. For the contrast experiment (*Figure 5—figure supplement 7*), the blocks of interest contained the low and high contrast stimuli, and the preceding fixation blocks served as the baseline. The time course in each epoch was converted to percent signal change by subtracting and then dividing by the baseline, and multiplying by 100. Converted time courses were then averaged across epochs, hemispheres in each run, and runs in each subject. The average signal change from 8 to 12 s after the onset of the block (the response peak) served as the measure of the fMRI response.

## MR spectroscopy

Spectroscopy data were acquired using a MEGA-PRESS (*Mescher et al., 1998*) sequence (3 cm isotropic voxel, 320 averages of 2048 data points, 2 kHz spectral width, 1.4 kHz bandwidth refocusing pulse, VAPOR water suppression, 2 s TR, 68 ms TE). Editing pulses (14 ms) were applied at 1.9 ppm during 'on' and 7.5 ppm during 'off' acquisitions, interleaved every 2 TRs across a 16-step phase cycle. MRS data were collected within the following regions (*Figure 5—figure supplement 1*): hMT+ in lateral occipital cortex, EVC in mid-occipital cortex, and a region of the central sulcus in parietal cortex known as the 'hand knob' (*Yousry et al., 1997*). Voxels were positioned based on anatomical landmarks using a $T_1$-weighted anatomical scan collected in the same session, while avoiding contamination by CSF, bone, and fat. The EVC voxel was placed medially within occipital cortex adjacent to the occipital pole, and aligned parallel to the cerebellar tentorium. The parietal voxel was centered on the 'hand knob' within the central sulcus, and aligned parallel to the dorsolateral cortical surface. The hMT+ voxel was placed in the ventrolateral occipital lobe, parallel to the lateral cortical surface. Further positioning information for hMT+ was obtained using an abbreviated version of the hMT+ fMRI localizer described above (65 TRs, each 3 s, resolution $3 \times 3 \times 5$ mm, 14 slices with 0.5 mm gap). These localizer data were processed on-line at the scanner, using a GLM analysis in the Philips iViewBOLD software to identify hMT+ voxels in lateral occipital cortex that responded more strongly to moving vs. static gratings (threshold $t > 3.0$). The hMT+ MRS voxels were centered on these functionally identified regions in the left and right hemispheres within each subject. To mitigate the detrimental effects of gradient heating during fMRI on the MRS data quality, the functional localizer data were acquired prior to the $T_1$ anatomical scan. MRS data were acquired in both left and right hMT+ for all subjects. Two measurements in EVC were obtained for all subjects except 1. Both values were averaged for hMT+ and EVC. Parietal cortex was measured only once, and was not obtained in one subject.

To quantify the concentration of GABA+ within each voxel, MRS data were processed using the Gannet 2.0 toolbox (*Edden et al., 2014*). We refer to this measurement as GABA+ to note that it includes some contribution from macromolecules that is not accounted for in our analysis (*Mullins et al., 2014*). Processing included automatic frequency and phase correction, artifact rejection (frequency correction parameters >3 *SD* above mean), and exponential line broadening (3 Hz). The GABA+ peak was fit with a Gaussian (*Figure 5—figure supplement 1D and E*), and the integral of the fit served as the concentration measurement. This GABA+ value was scaled by the integral of the unsuppressed water peak, fit with a mixed Gaussian-Lorentzian. The GABA+ value was corrected based on the concentration of gray and white matter using *Equation 2* from (*Harris et al., 2015*), assuming a ratio of GABA+ in white to gray matter ($\alpha$) of 0.5. Gray and white matter concentrations were obtained for each voxel in each subject by segmenting the $T_1$ anatomical data using SPM8 (*Friston et al., 1994*). This correction did not qualitatively affect our results. All MRS scans and corresponding psychophysical and fMRI data were collected within a maximum 2-week time-period, as previous work has shown GABA measurements are relatively stable across several days (*Evans et al., 2010*; *Greenhouse et al., 2016*).

## Statistics

All statistical analyses were performed in MATLAB. *F*-test statistics were obtained using repeated measures ANOVAs (e.g. four repeated psychophysical threshold estimates per subject), with

subjects treated as a random effect. Stimulus size was treated as a continuous variable where appropriate. To examine whether our data were normally distributed (as assumed by parametric tests such as an ANOVA), we manually inspected the distributions of our data in each condition, and used the Shapiro-Wilk test of normality. Cases in which deviations from normality were observed were further tested using the non-parametric equivalent of an ANOVA (Friedman's test). The p-values obtained in all such cases were smaller for the Friedman's test than for the corresponding ANOVA, thus we report the larger values and the more conventional statistic. Correlation values ($r$) are Pearson's correlation coefficients (two-tailed, unless otherwise noted). Significant correlations were confirmed using a (non-parametric) permutation test, which involved randomly shuffling the data being correlated across subjects in each of 10,000 iterations. The proportion of permuted correlations whose absolute value was greater than that of the real correlation served as the measure of significance. Permutation tests consistently yielded smaller p-values than the corresponding Pearson's correlations; the larger values are reported. Post-hoc power analyses were performed to ensure that sample sizes were large enough such that the probability of type II error was less than 20%. Importantly, the sample sizes in our correlational analyses (N = 21 to 27) give us enough power to detect moderate to strong correlations (Pearson's coefficients above $r$ = 0.58 to 0.52, depending on N) (*Hulley et al., 2013*), assuming a two-tailed significance threshold of p=0.05.

## Data availability

Data from this study are available from the Dryad Digital Repository: http://dx.doi.org/10.5061/dryad.rv71c

## Acknowledgements

We thank Geoffrey M. Boynton for providing the MATLAB functions for the model, and for comments on the manuscript. We also thank Brenna Boyd, Judy Han, Heena Panjwani, Micah Pepper, Meaghan Thompson, Anne Wolken, the UW Diagnostic Imaging Center, and the UW Investigational Drug Service for their help with subject recruitment and/or data collection. Finally, we thank the reviewers for their helpful comments. This work was supported by funding from the National Institute of Health (F32 EY025121 to MPS, R01 MH106520 to SOM, T32 EY007031, P41 EB015909 and R01 EB016089).

## Additional information

### Funding

| Funder | Grant reference number | Author |
| --- | --- | --- |
| National Eye Institute | F32 EY025121 | Michael-Paul Schallmo<br>Scott O. Murray |
| National Institute of Mental Health | R01 MH106520 | Raphael A Bernier<br>Scott O. Murray |
| National Institute of Biomedical Imaging and Bioengineering | P41 EB015909 | Richard AE Edden |
| National Eye Institute | T32 EY007031 | Michael-Paul Schallmo<br>Scott O. Murray |
| National Institute of Biomedical Imaging and Bioengineering | R01 EB016089 | Richard AE Edden |

The funders had no role in study design, data collection and interpretation, or the decision to submit the work for publication.

### Author contributions

Michael-Paul Schallmo, Conceptualization, Software, Formal analysis, Supervision, Funding acquisition, Investigation, Visualization, Methodology, Writing—original draft, Writing—review and editing;

Alexander M Kale, Software, Investigation, Methodology, Writing—review and editing; Rachel Millin, Investigation, Methodology, Writing—review and editing; Anastasia V Flevaris, Conceptualization, Software, Formal analysis, Investigation, Methodology, Writing—review and editing; Zoran Brkanac, Resources, Methodology, Writing—review and editing; Richard AE Edden, Resources, Software, Funding acquisition, Methodology, Writing—review and editing; Raphael A Bernier, Conceptualization, Resources, Supervision, Funding acquisition, Writing—review and editing; Scott O Murray, Conceptualization, Resources, Software, Formal analysis, Supervision, Funding acquisition, Methodology, Writing—original draft, Writing—review and editing

### Author ORCIDs
Michael-Paul Schallmo http://orcid.org/0000-0001-8252-8607
Alexander M Kale http://orcid.org/0000-0001-7668-2800

### Ethics

Human subjects: Subjects provided written informed consent prior to participation and were compensated for their time. All experimental procedures were approved by the University of Washington Institutional Review Board (protocol #s: 556, 1678, 28148), and conformed to the ethical principles for research on human subjects from the Declaration of Helsinki.

### Decision letter and Author response
Decision letter https://doi.org/10.7554/eLife.30334.021
Author response https://doi.org/10.7554/eLife.30334.022

## Additional files

### Supplementary files
• Transparent reporting form
DOI: https://doi.org/10.7554/eLife.30334.016

### Major datasets
The following dataset was generated:

| Author(s) | Year | Dataset title | Dataset URL | Database, license, and accessibility information |
| --- | --- | --- | --- | --- |
| Schallmo M-P, Kale AM, Millin R, Flevaris AV, Brkanac Z, Edden RAE, Bernier RA, Murray SO | 2018 | Data from: Suppression and facilitation of human neural responses | http://dx.doi.org/10.5061/dryad.rv71c | Available at Dryad Digital Repository under a CC0 Public Domain Dedication |

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

# Appendix 1

DOI: https://doi.org/10.7554/eLife.30334.017

In the Results and Materials and methods, we present a summary of the standard normalization model (*Equation 1*), which is a direct application of the work from *Reynolds and Heeger (2009)*. A more complete description is provided below. The parameters $E$ and $S$ represent the excitatory and suppressive drive within the model, which are a function of the spatial extent ($x$), orientation ($\theta$), and contrast ($c$) of the stimulus, as well as the width of model tuning in space and orientation for both excitation ($x_{w\_e}$, $\theta_{w\_e}$) and suppression ($x_{w\_s}$, $\theta_{w\_s}$). This relationship can be expressed as:

$$E(x,\theta,c) = e(x_{w\_e},\theta_{w\_e}) * N(x,\theta,c) \tag{A1}$$

$$S(x,\theta,c) = s(x_{w\_s},\theta_{w\_s}) * E(x,\theta,c) \tag{A2}$$

where $N$ is a 'neural image' that represents the population response to a given stimulus as a 2-dimensional Gaussian ($x$ and $\theta$), whose amplitude is set by $c$, and * denotes convolution. The terms $e$ and $s$ are also 2-D Gaussians that represent the selectivity (tuning width) of excitation and suppression, respectively. To predict the model response rate ($R$), the excitatory drive ($E$) is divided by the sum of the suppressive drive ($S$) and the semi-saturation constant ($\sigma$), as in *Equation 1*, which is reprinted here:

$$R = \frac{E}{S + \sigma} \tag{A3}$$

(same as *Equation 1*)

The predicted threshold ($T$) is determined according to the following procedure, which is summarized in *Equation 3*:

$$T = \frac{C}{R_p} \tag{A4}$$

(an extension of *Equation 3*)

where

$$R_p = \max(R) \tag{A5}$$

Thus, the threshold duration for motion discrimination predicted by the model ($T$) is a function of the peak response rate ($R_p$) and the criterion response level ($C$) required for a perceptual judgment. We use a winner-take-all rule (*Equation A5*) in determining $T$, which is consistent

with studies in macaques showing that behavioral performance during motion direction discrimination may be accounted for by the response of a small pool of neurons whose tuning properties are well-matched to the stimulus (i.e., centered in space and orientation) (*Liu et al., 2016*; *Britten et al., 1992*).

Full parameters for each application of the model are given below in *Appendix 1—table 1*. In the version of the model shown in *Figure 4—figure supplement 1* that characterizes the results from the experiment using lorazepam, we include an additional parameter *A* that scales the response *R* from *Equation A3*.

**Appendix 1—table 1.** Normalization model parameters. Arbitrary units abbreviated as a.u. Lorazepam abbreviated LZ. [+]Indicates parameters that were adjusted in order to fit the data, as opposed to those that were fixed to match the stimuli. Changes in contrast gain were modeled by varying stimulus contrast (e.g. stimulus contrast for the LZ model is the square root of the contrast for the placebo model). Response gain changes were modeled through the inclusion of a parameter that scaled the predicted response (*A*). For the MRS model, the effect of hMT+ GABA was modeled by varying the response criteria (lower value is 80% of the higher criterion).

| Parameter name | Parameter value | | |
|---|---|---|---|
| | **Basic model** | **Lorazepam model** | **hMT+ GABA MRS model** |
| Stimulus contrast | 0.03 or 0.98 | 0.03 or 0.98 (placebo), 0.017 or 0.099 (LZ) | 0.03 or 0.98 |
| Stimulus spatial center ($x$, a.u.) | 0 | 0 | 0 |
| Stimulus spatial width (a.u.) | 1, 2, or 12 | 0.84, 1.7, or 10 | 0.84, 1.7, or 10 |
| Stimulus orientation ($\theta$, °) | 90 | 90 | 90 |
| Stimulus orientation width (°) | 5 | 5 | 5 |
| Excitatory spatial pooling width ($x_{w\_e}$, a.u.)[+] | 5 | 6.5 | 4.5 |
| Excitatory orientation pooling width ($\theta_{w\_e}$, °)[+] | 25 | 5 | 25 |
| Suppressive spatial pooling width ($x_{w\_s}$, a.u.)[+] | 40 | 40 | 15 |
| Suppressive orientation pooling width ($\theta_{w\_s}$, °)[+] | 50 | 25 | 50 |
| Semi-saturation constant ($\sigma$, a.u.)[+] | 0.0002 | 0.0001 | 0.0001 |
| Criterion (at 3% contrast, a.u.)[+] | 300 | 775 | 300 (higher), 240 (lower) |
| Criterion (at 98% contrast, a.u.)[+] | 650 | 775 | 375 (higher), 300 (lower) |
| Response scalar ($A$, a.u.)[+] | N/A | 1 (placebo), 0.85 (LZ) | N/A |

DOI: https://doi.org/10.7554/eLife.30334.018

