## [Decision Letter]

Thank you for submitting your article "Suppression and facilitation of human neural responses" for consideration by *eLife*. Your article has been reviewed by three peer reviewers, and the evaluation has been overseen by Nick Turk-Browne as the Reviewing Editor and David Van Essen as the Senior Editor. The following individual involved in review of your submission has agreed to reveal his identity: Michael Silver.

The reviewers have discussed the reviews with one another and the Reviewing Editor has drafted this decision to help you prepare a revised submission.

Summary:

This authors use a variety of converging methods, including behavior, neuroimaging, spectroscopy, and pharmacology to provide a new account of visual motion perception. They conclude that the same mechanism may be responsible for both suppressive and facilitatory processes, and that neural inhibition may not underlie suppression. The reviewers all thought this was an impressive body of work, but they also raised several questions and concerns detailed below. If these constructive suggestions are addressed in a revision, we are optimistic and supportive about this work appearing in *eLife*.

Essential revisions:

1) One of the claims in the paper is that one computational mechanism can account for both suppression and facilitation, but the authors use different criterion parameters for low and high contrast stimulus (conditions where facilitation and suppression dominate, respectively). This difference is about two-fold, indicating that about twice as much stimulus response is needed to reach the threshold at high rather than at low contrast. This does require an explanation and/or justification. The criterion model implicitly assumes some temporal accumulation to a criterion that determines a threshold. Why would that differ between low and high contrast? More importantly, how important is this criterion difference for the authors' claim that the same model can explain both low and high contrast data? This is less of an issue if this is not critical for the authors' claim.

2) Statistical power and conclusions of MRS study. Only 22 subjects were included, which seems pretty small for an individual differences study. There is a statement about power analyses performed, but I was confused about what data the power analyses were performed on. Was this an a priori power analysis (from what dataset?) estimating that N=21 to 27 would be enough? Or was this a post-hoc power analysis on the actual dataset, which should only tell you the effect size? More information is needed if this is to be convincing. This issue is particularly problematic because the main finding is a null result; that is, a lack of a significant correlation. But the effect goes in the predicted direction, with r=-.32 and p=. 14 for high-contrast suppression, and the binary split data (Figure 5) also show something small in the predicted direction. It is one thing to state that there was not a statistically reliable result, but to definitively conclude a lack of effect, especially given the sample size, seems unwarranted (e.g. Abstract: "suppression is not driven by GABA-mediated inhibition"). Without this effect the conclusions and impact of the paper are certainly dampened, though it may still be sufficiently interesting to frame it as raising doubt about the GABA-mediated mechanism.

3) Impact of computational modeling study. The authors explain in the methods: "We emphasize that our goal was not to quantitatively describe the data in terms of the model but instead to show that normalization, as a computational principle, is sufficient to qualitatively account for our findings." I find this a bit unsatisfying. Does it really tell us much to say that a particular model could account for the behavioral pattern? (1) I don't think that this "could" statement is particularly surprising – divisive normalization is a broad, pretty mainstream model of lots of things now. (2) If divisive normalization can account for the pattern, but a 2-process model could account for it better, the conclusion would be completely different. I think there's some merit in including the modeling, but if there's not going to be a quantitative comparison, the interpretation and weighting of this part of the paper should be toned down.

4) fMRI study design. I was puzzled by several of the fMRI design and analysis decisions. I would have assumed that a standard block design to test this question would include blocks of all condition types within a single run, with counterbalanced order, and some fixation blocks interspersed. Instead they only tested 2 conditions per run, in strictly alternating order, with no fixation period between. Then the analysis was to simply take the average raw BOLD signal over the timecourse of a given type of block, with a "baseline signal" subtracted. But the baseline was taken 0-4s prior to block onset, which corresponds to the peak of the previous block (which was always the opposite condition). Given this, why take a baseline at all, rather than just directly compare the conditions? These choices produce some serious interpretation concerns if I'm understanding their design/analysis correctly.

5) One factor that surely affects the authors' fMRI results are different RF sizes in MT and EVC. This matters as the stimulus sizes (small, medium and large) are fixed, so the relative differences between stimulus size and average RF size will differ between MT and EVC. This will surely affect any signal that depends on a spatial activation of visual RFs. This is a concern that really limits a comparison between EVC and MT results-a limitation that needs to be explicitly addressed.

[Editors' note: further revisions were requested prior to acceptance, as described below.]

Thank you for resubmitting your work entitled "Suppression and facilitation of human neural responses" for further consideration at *eLife*. Your revised article has been favorably evaluated by David Van Essen (Senior editor), Nick Turk-Browne (Reviewing editor), and two of the original reviewers.

The manuscript has been improved but there are some remaining issues that need to be addressed before acceptance, as outlined below:

1) There is still ambiguity about the fMRI design. In the rebuttal, the authors clarify that in the alternating block design, the small stimulus represents the "baseline" and the question is whether there is any change from this baseline during the large stimulus blocks. However, this is not how the analysis is described in the Materials and methods. The Materials and methods still state that "For each type of block (e.g. small and large stimuli), response baseline was determined by averaging the signal across all such epochs between 0-4 s prior to block onset." This makes it seem like there are 2 conditions of interest in each run (small, large), and each is compared to its own baseline. This was the source of initial confusion, because "baselines" here are confounded with the other condition. From the rebuttal, though, it sounds like each run has a single condition of interest ("large"), and this is compared to the baseline of "small". This would explain why the figures only show small-to-large timecourses and not also large-to-small. The Materials and methods need to be revised to be much more clear about this.

2) In the rebuttal, the authors explicitly clarify that "we can say there is no evidence for strong, or even moderate correlations (stronger than r =.52) in our data […]". This statement/caveat should be made explicit in the manuscript, and not just in the Materials and methods; this seems like an important Discussion point. In particular, what are the theoretical implications if there were a weak but reliable correlation between these factors? The current experiment with this power can only detect strong correlations – but does the absence of a strong/moderate correlation imply that spatial suppression "is not directly mediated" by inhibition? Is there a theoretical reason to think that a hypothetical weak but reliable correlation would be evidence of something other than direct mediation? What is the a priori threshold needed to claim "direct mediation"? The authors need to address (in the manuscript) the fact that they only have the power to detect strong correlations, why they think it's theoretically justified to only focus on strong/moderate correlations, and leave open the possibility that there may still be weaker correlations that could not be detected here. Along these lines, the authors did not sufficiently "soften" their claims. In addition to the lack of caveats or softening in the Discussion, the results still make statements such as "no correlations were found", "there was no association between", and "no relationship between".

3) In the rebuttal and revision the authors are claiming that the data in Figure 5 if anything show an effect in the opposite direction as predicted, but this is confusing. When looking at Figure 5, there seems to be numerically greater suppression for higher GABA (i.e. Figure 5, the size index for high contrast s-b is more negative for high GABA than low GABA). Whether this is driven by a change in the small vs large duration thresholds might be unexpected, but it is still in the direction of an increase in "suppression", as defined as this size index and used throughout the rest of the paper. Again, not suggesting to make anything of this effect – it is clearly not statistically reliable in this sample – but it remains possible than an effect could emerge in a larger sample size, and this can't be so readily dismissed as a clear absence of an effect.

---

## [Author Response]

Essential revisions:1) One of the claims in the paper is that one computational mechanism can account for both suppression and facilitation, but the authors use different criterion parameters for low and high contrast stimulus (conditions where facilitation and suppression dominate, respectively). This difference is about two-fold, indicating that about twice as much stimulus response is needed to reach the threshold at high rather than at low contrast. This does require an explanation and/or justification. The criterion model implicitly assumes some temporal accumulation to a criterion that determines a threshold. Why would that differ between low and high contrast? More importantly, how important is this criterion difference for the authors' claim that the same model can explain both low and high contrast data? This is less of an issue if this is not critical for the authors' claim.

The reviewers raise an interesting and important point, and we have provided additional explanation and justification for our use of different criterion values within the model for low and high contrast stimuli. These additions appear in the Computational Modeling section of the Materials and methods. Our additions to the manuscript attempt to capture the following points:

First, we want to emphasize that a single criterion is all that is required to account for the principle duration-threshold findings across all similar published experiments: suppression for high contrast stimuli and summation (and/or summation followed by suppression) for low contrast stimuli. So, in this sense, using different criterion values for different contrast values is not necessary for the model to explain the main characteristics of duration threshold experiments. Also, we note that the model describing the results from the LZ experiment uses a single criterion for both low and high contrast stimuli (Figure 4—figure supplement 1).

However, there is a particular characteristic of duration threshold findings (present in many publications, including the original Tadin, 2003 Nature paper and in some of our data in this manuscript) that exposes a limitation of the linking hypothesis that thresholds are directly proportional to neural responses in MT (or any other early visual area). Specifically, duration thresholds can be equivalent or shorter for low contrast stimuli compared to high contrast stimuli (this can be seen in our data, Figure 2, medium and large sizes). If we fully accept the link between MT responses and duration thresholds, this pattern of behavior would predict that MT responses are greater for very low (3%) contrast stimuli than for very high (98%) contrast stimuli. Responses in MT, however, are strongly driven by luminance contrast (a well-established finding in the field). To further demonstrate this, we have added additional fMRI measurements showing a large increase in MT and EVC responses with increasing luminance contrast (Figure 5—figure supplement 7). This leads us to the conclusion is that there must be an additional factor (outside of MT and EVC response magnitude) that slows the evidence accumulation rate at high contrast and speeds it up at low contrast. One plausible explanation for slower accumulation at high contrast is stimulus onset masking. Masking has been shown to contribute to the spatial suppression effect, and is thought to be stronger for high contrast stimuli (Churan, 2009; Tsui, 2011).

From a modeling perspective, we addressed this issue by including different criterion values for low and high contrast stimuli in Equation 3, which transforms the ‘MT model’ response to a ‘behavioral’ threshold – it is basically a way for us to simplistically acknowledge within the model that there are ‘downstream’ contrast-dependent processes (e.g., masking) that are beyond the scope of the current manuscript. Further research will be needed to clarify how factors such as masking contribute to spatial suppression, and how this may be described within a model framework.

2) Statistical power and conclusions of MRS study. Only 22 subjects were included, which seems pretty small for an individual differences study. There is a statement about power analyses performed, but I was confused about what data the power analyses were performed on. Was this an a priori power analysis (from what dataset?) estimating that N=21 to 27 would be enough? Or was this a post-hoc power analysis on the actual dataset, which should only tell you the effect size? More information is needed if this is to be convincing. This issue is particularly problematic because the main finding is a null result; that is, a lack of a significant correlation. But the effect goes in the predicted direction, with r=-.32 and p=. 14 for high-contrast suppression, and the binary split data (Figure 5) also show something small in the predicted direction. It is one thing to state that there was not a statistically reliable result, but to definitively conclude a lack of effect, especially given the sample size, seems unwarranted (e.g. Abstract: "suppression is not driven by GABA-mediated inhibition"). Without this effect the conclusions and impact of the paper are certainly dampened, though it may still be sufficiently interesting to frame it as raising doubt about the GABA-mediated mechanism.

We appreciate the reviewers’ concern, and have taken steps to address this issue. First, the power analysis was indeed post-hoc, and was used to answer the question “What is the smallest correlation that we have sufficient power to detect, given our sample size?” Based on a standard type-II error rate of 0.2, we determined that our sample sizes (n = 21 to 27, depending on the analysis) gave us sufficient power to detect correlations of r ≥ 0.52, should such correlations exist. The reviewers are right to state that we cannot claim that there’s no correlation between GABA and suppression exists on the basis of these data. Rather, we can say that there is no evidence for strong, or even moderate correlations (stronger than r = 0.52) in our data – correlations that we would have sufficient power to detect in our sample if they existed.

We have clarified the fact that our data in Figure 5 (median split by GABA+ in hMT+) did not show the predicted pattern of higher thresholds for large stimuli (stronger suppression) with greater GABA+; if anything, there is a greater effect of GABA+ for small high contrast stimuli, which does not match our prediction. It is also worth noting that with 8 correlations for the psychophysical and fMRI suppression metrics, we would expect to find a p-value < 0.12 just by chance alone (without correction for multiple comparisons). Further, the observed correlation coefficients do not follow a consistent pattern across stimulus conditions, suggesting no consistent relationship between GABA concentration and suppression magnitude.

Nevertheless, we agree that the portions of the text that discuss this finding should be softened. We have taken steps to adjust our statements regarding the MRS results and overall role of GABA in suppression throughout the manuscript.

3) Impact of computational modeling study. The authors explain in the Materials and methods: "We emphasize that our goal was not to quantitatively describe the data in terms of the model but instead to show that normalization, as a computational principle, is sufficient to qualitatively account for our findings." I find this a bit unsatisfying. Does it really tell us much to say that a particular model could account for the behavioral pattern? (1) I don't think that this "could" statement is particularly surprising – divisive normalization is a broad, pretty mainstream model of lots of things now. (2) If divisive normalization can account for the pattern, but a 2-process model could account for it better, the conclusion would be completely different. I think there's some merit in including the modeling, but if there's not going to be a quantitative comparison, the interpretation and weighting of this part of the paper should be toned down.

We have revised this section to provide greater clarity on the goals of the modeling work, and toned down the interpretation and weight according to the reviewer’s suggestion. While we do provide a quantitative description of our results in terms of the model parameters, our intention with this work was not to find the precise parameter values that provide the best algorithmic fit to our data. Rather, we sought to demonstrate that, using a reasonable set of manually derived parameters, a well-accepted single process model (divisive normalization) is sufficient to explain both suppression and summation. This initial effort provided proof-of-concept; future experiments with a larger number of stimulus conditions (i.e., sizes and contrasts), will permit more precise fitting of model parameters and direct comparisons with 2-stage models. This will be of particular interest when applying this paradigm within clinical populations (e.g., does a change within a single parameter of this model account for the behavior observed in a given population?)

4) fMRI study design. I was puzzled by several of the fMRI design and analysis decisions. I would have assumed that a standard block design to test this question would include blocks of all condition types within a single run, with counterbalanced order, and some fixation blocks interspersed. Instead they only tested 2 conditions per run, in strictly alternating order, with no fixation period between. Then the analysis was to simply take the average raw BOLD signal over the timecourse of a given type of block, with a "baseline signal" subtracted. But the baseline was taken 0-4s prior to block onset, which corresponds to the peak of the previous block (which was always the opposite condition). Given this, why take a baseline at all, rather than just directly compare the conditions? These choices produce some serious interpretation concerns if I'm understanding their design/analysis correctly.

The reviewer is right to point out the importance of our experimental design in measuring suppression and summation using fMRI. We have provided additional citations and greater detail in both the Results and Materials and methods sections regarding this aspect of the experiment, in order to clarify this point.

We note that this alternating-block experimental design has a long history of use for measuring visual surround suppression via fMRI (e.g., Zenger-Landolt and Heeger, 2003; Williams and Singh, 2003). By first defining an ROI that responds selectively to the smaller stimulus (center > surround; Olman and Heeger, 2007; Schallmo and Olman, 2016), and comparing the fMRI response within this ROI between small (our baseline) and large (suppression or summation) blocks, we established a null hypothesis of no change in the fMRI response. That is, if the surrounding stimuli had no effect on the response to the center, then the response should not change between blocks, as the center stimulus was constant across both. Essentially, we are directly comparing the relative fMRI responses to small and large stimuli, without referencing these responses to a separate (potentially noisy) baseline response to fixation alone.

While mixed block designs have also been used during fMRI to study surround suppression (e.g., Pihlaja, 2008), the current paradigm was designed for (ongoing) use within a clinical population, and thus is intended to be especially robust to subject noise (e.g., head motion). By not relying on a GLM, this paradigm permits us to determine whether there were any outliers in the fMRI response on a block-by-block basis, and exclude them (a step not required in the current study).

5) One factor that surely affects the authors' fMRI results are different RF sizes in MT and EVC. This matters as the stimulus sizes (small, medium and large) are fixed, so the relative differences between stimulus size and average RF size will differ between MT and EVC. This will surely affect any signal that depends on a spatial activation of visual RFs. This is a concern that really limits a comparison between EVC and MT results-a limitation that needs to be explicitly addressed.

We agree with the reviewer’s point, and have greatly expanded upon the existing consideration of this issue (from the Results) in the Discussion section. By first localizing hMT+, and then finding voxels within this ROI that respond preferentially to center > surround (from a separate localizer scan), we sought to identify the most comparable voxels to those identified in EVC, within the limitations of the current fMRI technique. We now explicitly discuss this limitation of our method as it relates to comparing results between hMT+ and EVC.

[Editors' note: further revisions were requested prior to acceptance, as described below.]The manuscript has been improved but there are some remaining issues that need to be addressed before acceptance, as outlined below:1) There is still ambiguity about the fMRI design. In the rebuttal, the authors clarify that in the alternating block design, the small stimulus represents the "baseline" and the question is whether there is any change from this baseline during the large stimulus blocks. However, this is not how the analysis is described in the Materials and methods. The Materials and methods still state that "For each type of block (e.g. small and large stimuli), response baseline was determined by averaging the signal across all such epochs between 0-4 s prior to block onset." This makes it seem like there are 2 conditions of interest in each run (small, large), and each is compared to its own baseline. This was the source of initial confusion, because "baselines" here are confounded with the other condition. From the rebuttal, though, it sounds like each run has a single condition of interest ("large"), and this is compared to the baseline of "small". This would explain why the figures only show small-to-large timecourses and not also large-to-small. The Materials and methods need to be revised to be much more clear about this.

We thank the reviewers for pointing out the lack of clarity in this section, and have taken steps to improve our explanation of the fMRI analysis. In particular, we now explicitly point out in both the Results and Materials and methods that: 1) the large stimulus block was our “block of interest,” and 2) that the response from the end of the small stimulus block served as the response baseline.

2) In the rebuttal, the authors explicitly clarify that "we can say there is no evidence for strong, or even moderate correlations (stronger than r =.52) in our data […]". This statement/caveat should be made explicit in the manuscript, and not just in the Materials and methods; this seems like an important Discussion point. In particular, what are the theoretical implications if there were a weak but reliable correlation between these factors? The current experiment with this power can only detect strong correlations – but does the absence of a strong/moderate correlation imply that spatial suppression "is not directly mediated" by inhibition? Is there a theoretical reason to think that a hypothetical weak but reliable correlation would be evidence of something other than direct mediation? What is the a priori threshold needed to claim "direct mediation"? The authors need to address (in the manuscript) the fact that they only have the power to detect strong correlations, why they think it's theoretically justified to only focus on strong/moderate correlations, and leave open the possibility that there may still be weaker correlations that could not be detected here. Along these lines, the authors did not sufficiently "soften" their claims. In addition to the lack of caveats or softening in the Discussion, the results still make statements such as "no correlations were found", "there was no association between", and "no relationship between".

We have expanded upon our discussion of this issue to include the statement regarding “no evidence for strong or even moderate correlations […]” in the second paragraph of the Discussion. We now explicitly mention the possibility of weak, undetected correlations between GABA+ and suppression strength, and the implications of such. Further, we have taken greater measures to soften the language surrounding this point, both here in the Discussion, as well as in the MRS Results (as noted by the reviewers), and at the end of the Introduction.

3) In the rebuttal and revision the authors are claiming that the data in Figure 5 if anything show an effect in the opposite direction as predicted, but this is confusing. When looking at Figure 5, there seems to be numerically greater suppression for higher GABA (i.e. Figure 5, the size index for high contrast s-b is more negative for high GABA than low GABA). Whether this is driven by a change in the small vs large duration thresholds might be unexpected, but it is still in the direction of an increase in "suppression", as defined as this size index and used throughout the rest of the paper. Again, not suggesting to make anything of this effect – it is clearly not statistically reliable in this sample – but it remains possible than an effect could emerge in a larger sample size, and this can't be so readily dismissed as a clear absence of an effect.

This observation is now directly addressed in the MRS Results section – we point out the numerically greater suppression in the high GABA+ group, note the lack of statistical significance in our sample, and point out the difference in thresholds for small stimuli as the likely source of this trend.